# Thioredoxin-1 and Correlations of the Plasma Cytokines Regarding Aortic Valve Stenosis Severity

**DOI:** 10.3390/biomedicines9081041

**Published:** 2021-08-18

**Authors:** Peteris Tretjakovs, Juris Lurins, Simons Svirskis, Gita Gersone, Dace Lurina, Ulla Rozenberga, Leons Blumfelds, Guntis Bahs, Aivars Lejnieks, Vitolds Mackevics

**Affiliations:** Faculty of Medicine, Riga Stradins University, 16 Dzirciema Str., LV-1007 Riga, Latvia; Cor7@inbox.lv (J.L.); Simons.Svirskis@rsu.lv (S.S.); Gita.Gersone@rsu.lv (G.G.); dace.kardio@gmail.com (D.L.); ulla.rozenberga@gmail.com (U.R.); Leons.Blumfelds@rsu.lv (L.B.); Guntis.Bahs@rsu.lv (G.B.); Aivars.Lejnieks@rsu.lv (A.L.); Vitolds.Mackevics@rsu.lv (V.M.)

**Keywords:** aortic valve stenosis, thioredoxin-1, myeloperoxidase, growth factors, cytokines

## Abstract

Aortic valve stenosis (AS) develops not only with a pronounced local inflammatory response, but also oxidative stress is involved. The aim of this study was to evaluate the plasma levels of thioredoxin-1 (TRX1), myeloperoxidase (MPO), chemerin, growth differentiation factor 15 (GDF-15), angiopoietin-2 (Ang-2), vascular endothelial growth factor A (VEGF-A), fibroblast growth factor 2 (FGF-2), fibroblast growth factor 21 (FGF-21), and metalloproteinase (MMP)-1, -3, and -9 in acquired AS patients as well as to clarify the correlations of TXR1 and the plasma inflammatory biomarkers regarding AS severity. AS patients were classified into three groups: 16 patients with mild AS stenosis, 19 with moderate and 11 with severe AS, and 30 subjects without AS were selected as a control group. AS patients had significantly higher plasma levels of TRX1 compared to controls, but the highest difference was found in mild AS patients compared to the controls. We conclude that AS is associated with significantly increased plasma TRX1 levels, and TRX1 might serve as a specific and sensitive biomarker of AS. TRX1 and also chemerin, GDF-15, VEGF-A, FGF-2 and FGF-21 significantly correlate with AS severity degrees. TRX1 also showed positive association with FGF-2, VEGF-A, and MMP-3 in all AS patients.

## 1. Introduction

Oxidative stress plays an important role in the development of cardiovascular disorders [1,2]. The thioredoxin system is one of the main regulators for protection against oxidative stress. The system consists of thioredoxin-1 (TRX1), selenoenzyme thioredoxin reductase-1 (TrxR1) and NADPH and regulates the protein dithiol/disulphide balance [1]. The suppressive effect of TRX1, a redox-active low-molecular weight protein, on oxidative stress [3] is related to the cysteine pair of the active site of TRX1, resulting in the formation of a disulfide that can be reduced by TrxR1 using NADPH [4]. 

The thioredoxin system is an important antioxidant system that controls not only redox balance but also signal transduction in cells [4]. Reduced TRX1 is associated with various physiological functions, such as the inhibition of apoptosis or the control of the activity of many transcription factors [5]. At the same time, there is a thioredoxin system as one player and a glutathione and glutaredoxin system as another player in fighting against oxidative stress. Studies are currently being carried out into the possible use of the redoxins as drugs to combat oxidative stress-related diseases [6].

Another biomarker is myeloperoxidase (MPO), which is also associated with oxidative stress. MPO, a leukocyte-derived redox enzyme, catalyzes the formation of reactive oxygen species and is an index of oxidative stress [7]. It is interesting that MPO exerts effects that are beyond its oxidative properties, e.g., proinflammatory properties [8,9]. Several types of tissue injuries and chronic diseases, including cardiovascular diseases, are associated with MPO-derived oxidants [10]. Recent findings show that MPO is highly expressed in the aortic valve stenosis (AS) patients [11]. Other data suggest that endothelial dysfunction is mainly due to increased MPO release (causing inflammation) and has an impact primarily on the integrity of the valve rather than the aortic structure [12]. 

Oxidative stress reflects an imbalance between the excessive production of reactive oxygen species (ROS) and ability of antioxidant systems for ROS elimination. An imbalance in this protective mechanism (antioxidant defense) is related to the activation of proinflammatory genes, and it leads to the secretion of various cytokines and growth factors by different cells. Inflammation triggered by oxidative stress can be closely related to AS development [2,7,8,10].

A great number of inflammatory biomarkers, e.g., cytokines and growth factors, have been identified, with both diagnostic and prognostic relevance in AS patients with and without other cardiovascular diseases. Studies confirm that growth differentiation factor 15 (GDF-15), angiopoietin-2 (Ang-2), vascular endothelial growth factor (VEGF), and fibroblast growth factor 2 (FGF-2) are implicated in inflammation and angiogenesis [13,14,15]. Anti-inflammatory growth factor fibroblast growth factor 21 (FGF21) is also of clinical importance [16,17]. Metalloproteinase (MMP)-9 is involved in the pathogenesis of AS [18,19]. 

Studies have shown that the adipokine chemerin plays an important role in the development of cardiovascular diseases, including AS [20,21], and that chemerin can contribute to chronic inflammation and increased oxidative stress in obese individuals [22].

We hypothesized that TRX1, a cytosolic and extracellular enzyme with anti-oxidative, anti-apoptotic and anti-inflammatory properties [3], in AS patients is increased in plasma and, together with inflammatory cytokines, correlates with the severity of AS, and TRX1 may serve as a clinical biomarker.

The aim of this study was to evaluate the plasma levels of TRX1 and also MPO, chemerin and growth factors GDF-15, Ang-2, VEGF-A, FGF-2, FGF-21, MPP-1, -3, -9, and C-reactive protein (CRP) in acquired AS patients as well as to clarify the correlations of TXR1 and the plasma inflammatory biomarkers regarding AS severity.

## 2. Materials and Methods

### 2.1. Study Subjects

AS patients were allocated into three groups: 16 patients with mild AS stenosis; 19 with moderate and 11 with severe AS in line with the 2012 European Society of Cardiology and the European Association for Cardio-Thoracic Surgery Guidelines for the Management of Valvular Heart Disease [23]. Thirty subjects without AS (echocardiographically approved) were chosen as a control group. The study groups matched by age and body mass index.

Patients and controls were recruited according to the echocardiographically confirmed findings, and the data were obtained by applying GE VIVID 7 Dimension Cardiovascular Ultrasound system (GE Healthcare; GE Healthcare, Chicago, IL, USA) and Philips IE 33 Ultrasound Equipment (Philips Healthcare, Amsterdam, The Netherlands). Each EchoCG examination was performed by two professionals. 

Three groups of patients with AS were selected according to the severity grade in line with current guidelines on the management of valvular heart disease and EchoCG criteria: aortic jet velocity (Vmax) (m/s); mean pressure gradient, PG (mmHg); aortic valve area, AVA (cm^2^) and indexed AVA (cm^2^/m^2^). Data were graded as severe: Vmax > 4 m/s, PG > 40 mmHg, AVA < 1.0 cm^2^, indexed AVA < 0.6; moderate: Vmax 3.0–4.0 m/s, PG 20–40 mmHg, AVA 1.0–1.5 cm^2^, indexed AVA 0.60–0.85; mild: Vmax 2.5–2.9 m/s, PG < 20 mmHg, AVA > 1.5 cm^2^, indexed AVA > 0.85 [24].

The following exclusion criteria were applied for all of the study participants: pathologies of other valves and rheumatic aortic valve disease (by EchoCG, history of rheumatism); bicuspid aortic valve, cardiomyopathies, cardiac fibrosis, left ventricular systolic dysfunction (EF below 50%); atherosclerosis of coronary arteries; peripheral atherosclerosis (detecting intima-media thickness of carotid arteries, evaluating the ankle brachial index, history of peripheral artery disease, stroke); moderate, severe and uncontrolled arterial hypertension; obesity, diabetes mellitus, thyroid disfunction, smoking; connective tissue diseases, infectious diseases, oncological diseases; hypercholesterolemia and hypertriglyceridemia, including history of statin and fibrate use. 

The current study was acknowledged by the Riga Stradins University (Latvia) Ethics Committee on Research on Humans (No 12.09.2013/11). The study protocol agrees with the Ethical Guidelines of the 1975 Declaration of Helsinki, revised in 2008.

### 2.2. Laboratory Assays

Study subjects venous blood samples were collected after overnight fasting, centrifuged, and stored at −80 °C. TRX1 (Sigma-Aldrich Chemie GmbH, St. Louis, MO, USA, RAB1756-1KT), MPO (Cayman Chemical, Ann Arbor, MI, USA, 501410), Chemerin (Abcam, Cambridge, UK, ab155430), GDF-15 (Sigma-Aldrich Chemie GmbH, RAB0204-1KT), Ang-2 (Sigma-Aldrich Chemie GmbH, RAB0016-1KT), VEGF-A (Sig-ma-Aldrich Chemie GmbH, RAB0507-1KT), FGF-2 (Sigma-Aldrich Chemie GmbH, RAB0182-1KT), FGF-21 (Merck KGaA, Darmstadt, Germany, EZHFGF21-19K), MMP-1, -3, -9 (Merck KGaA, QIA55-1EA; RayBiotech Life, Inc., Peachtree Corners, GA, USA, ELH-MMP3-1-RB; RayBiotech Life, Inc., ELH-MMP9-001) and CRP (Sigma-Aldrich Chemie GmbH, RAB0096-1KT) were measured in plasma by ELISA method using TECAN Infinite 200 PRO multimode reader (Tecan Group, Ltd., Mannedorf, Switzerland). Concentrations of lipids, glucose, and other routine blood biomarkers were analyzed by standard methods.

### 2.3. Statistical Analysis 

The normal distribution of the data was proved by D’Agostino and Pearson, Anderson–Darling, and Shapiro–Wilk normality tests. The homogeneity of variances was tested using F-test (2 groups) or Brown–Forsythe and Bartlett’s tests (≥ 3 groups). Where data were non-normally distributed, non-parametric Mann–Whitney (MW) U-tests or Kruskal–Wallis (KW) H-tests, followed by the two-stage step-up method of Benjamini, Krieger, and Yekutieli as a post hoc procedure, were applied, and results were displayed as a median and interquartile range (IQR). When appropriate, Welch’s t test was applied and the mean with 95% confidential interval (CI) was displayed.

Correlation analysis was performed to find out the relationship between studied growth factors. Regression analysis was performed to determine whether changes in plasma growth factors concentrations were associated with AS severity. The *p* value of less than 0.05 (*p* < 0.05) was considered statistically significant for all used statistical tests.

The performance of the study biomarkers was assessed using receiver-operating characteristic (ROC) curves, sensitivity, specificity, and negative and positive predictive values. The *p*-value was reported for the area under the curve (AUC) for the best cut-off level. Diagnostic tests were assessed by this classification: 0.90–1 = excellent; 0.80–0.90 = good; 0.70–0.80 = fair; 0.60–0.70 = poor; and 0.50–0.60 = fail.

All graphical images and statistical analyses were performed using GraphPad Prism 9.0 for MacOS software (GraphPad Software, San Diego, CA, USA).

## 3. Results

### 3.1. Patient Characteristics and CRP Level Differences between the Patient Groups

The main data of the participants of the study are presented in Table 1. The average age of patients in the three aortic stenosis groups and the control group was relevant, and the mean body mass index (BMI) did not vary among the groups. The groups were equal for the mean values of the ejection fraction (EF) defined by Simpson’s method and the stroke volume index (SVI) assessed by the left ventricular outflow method and according to the inclusion and exclusion criteria. 

Patients with AS had a significantly higher level (*p* = 0.0107) of CRP compared to controls (Figure 1a), and the CRP level was higher only in mild AS (*p* = 0.0147) (Figure 1b). The correlation analysis revealed a positive relationship (r = 0.2732, *p* = 0.0065) between CRP levels and the control level (0) and degrees of AS severity (Figure 1c).

### 3.2. TRX1 Level Differences between the Patient Groups and ROC Analysis

Patients with AS had significantly higher levels (*p* = 0.0008) of TRX1 compared to controls (Figure 2a,b), but the ROC analysis showed that TRX1 cannot serve as a specific and sensitive biomarker for AS stenosis without grading the severity (poor level: AUC = 0.66, *p* = 0.0052) (Figure 2c).

The correlation analysis revealed a positive relationship (*p* = 0.0368) between TRX1 levels and the degree of AS severity (Figure 3).

Regarding the severity of AS, a statistically significant difference in TRX1 levels was found between control subjects and patients with mild and severe AS (*p* < 0.0001, *p* < 0.05), respectively, but not with moderate AS (Figure 4).

Our findings (by the ROC analysis) suggest that TRX1 might serve as a specific and sensitive biomarker only in mild AS patients (good level: AUC = 0.82, *p* = 0.0001) (Figure 5), but it did not serve as a specific and sensitive biomarker in patients with moderate to severe AS (AUC = 0.54, *p* = 0.66; AUC = 0.65, *p* = 0.08).

### 3.3. Correlations between the Study Biomarkers

The results of the correlation analysis reveal that AS patients have higher elevations in plasma concentrations of TRX1, FGF-2, VEGF-A, FGF-21, and especially GDF-15, while chemerin levels significantly correlate in AS patients compared to controls (Figure 6a). This relationship is also observed when the severity of AS is taken into account (Figure 7). 

To clarify how plasma levels of TRX1 correlate with the studied biomarkers, separate analysis for control and for AS group was performed, and since the distribution of the data in the respective subgroups was not uniform, to obtain more precise results, both parametric and non-parametric correlation analyses were performed. 

Thus, in control patients, a positive weak-to-moderate association between TRX1 and FGF-2, VEGF-A, FGF-21 and chemerin was shown, and a weak negative association was revealed between TRX1 and Ang-2, as well MMP1 (Figure 6b). More pronounced positive associations were observed in the AS patient group. To make more objective comparison and evaluate this relationship among control and AS subgroups, the distance between two correlation coefficients (one from each group of the respective biomarker) was calculated by applying expression Δr = r_max_ − r_min_. The results show that there are very positive associations between plasma TRX1 and MMP-3 and, in particular, FGF-2 levels in AS patients (Figure 6c). Spearman’s rank correlation analysis shows that CRP has some positive associations with TRX1 (r = 0.25), as well as with MMP-9 (r = 0.26), VEGF-A and FGF-21 (both r = 0.22), (Figure 7).

A moderately positive correlation (r = 0.3728, *p* = 0.0071) between MPO and MMP-9 levels was also detected in patients with AS (Figure 8).

## 4. Discussion

Development of degenerative aortic valve disease is no longer considered as a simple passive process of calcium deposition but as a complex, regulated process that also involves the extent of inflammatory activation and oxidative stress, especially in the early stages of AS, where it first causes endothelial dysfunction and then affects other aortic valve leaflet cell types [25,26,27]. 

In a previous study, we found that the plasma levels of inflammatory cytokines vary with the degree of AS, and we concluded that inflammatory chemokine chemerin is a good biomarker for the diagnosis of mild AS [21]. 

Concerning oxidative stress, the thioredoxin system is known to be one of the major antioxidant systems for protection against it [1]. This is consistent with the data from our study; we found an increase in TXR1 levels, especially in patients with mild AS when active inflammation predominates. Obviously, the oxidative stress factors are very pronounced and possibly also significant in this stage of pathogenesis. 

In addition to this finding, TRX1 showed significant correlation regarding AS severity degrees. It should be noted that in our study, many exclusion factors and diseases were taken into account in AS patients, as they are known to be associated with changes in cytokines and oxidative stress biomarkers. As with chemerin, TRX1 may be a specific and sensitive biomarker in patients with mild AS (at a good diagnostic level).

In another previous study, we also found that plasma concentrations of several growth factors are increased in AS patients [28]. Our results in this study show that not only TRX1, but also proinflammatory biomarkers, namely chemerin, GDF-15, VEGF-A, and FGF-2, and the anti-inflammation biomarker FGF-21 have significant correlations regarding AS severity degrees. Analyzing the correlations of TRX1 with other biomarkers explored in the study, we found that TRX1 was significantly correlated with FGF-2 and VEGF-A, but with MMP-3, it was less pronounced in AS patients. The results of this study suggest that patients with AS may have low-grade chronic inflammation and, interestingly, that CRP can correlate with TRX1.

Several studies confirm the clinical and diagnostic role of circulating GDF-15 in AS patients [29,30]. The angiogenic markers VEGF-A and FGF-2 are involved in matrix remodeling, and proliferation plays a significant pathophysiological role in the development of AS [31]. In turn, the anti-inflammatory cardiomyokine FGF-21’s effects on angiogenesis are associated with endothelial cells [16]. There are very few studies on the role of FGF-2 and VEGF in the pathogenesis of AS, although there is some evidence that FGF-2 activates valve interstitial cells (VIC) and that it can cause valve diseases such as fibrotic AS [31]. The structural remodeling of the valve cusp might reduce oxygen availability, contributing to the upregulation of VEGF and the formation of new blood vessels [32]. There is evidence that VEGF production was increased in the early stages of valve pathology, and, in turn, oxidized low-density lipoproteins stimulates myofibroblastic VIC differentiation and secretion of many inflammatory cytokines [33]. Based on our data, it can be hypothesized that TRX1, as a representative of the antioxidant system, may be involved in the structural remodeling of valves in the pathogenesis of AS.

Interestingly, we found a strong association between plasma MPO and MMP-9 levels in AS patients. MPO is linked to both inflammation and oxidative stress [34]. MPO-derived oxidants contribute to tissue damage in many diseases, especially those characterized by inflammation [9]. Many studies suggest that the enhanced level of MPO is one of the best diagnostic tools of inflammatory and oxidative stress biomarkers for cardiovascular disease [34]. MMP-9 can promote transmigration of monocytes across the endothelium. Macrophages and other cells can secrete MMP-9 in response to inflammatory biomarkers [35]. MMP-9 is involved in aortic valve structure modification in AS patients [18].

Oxidative stress can have a significant impact on the development of age-related diseases, including AS, as well as atherosclerosis and vascular disease. High concentrations of ROS can affect the progression of age-related diseases through oxidative damage, which is dependent on the inherited or acquired defects in enzymes involved in the redox-mediated signaling pathways [36].

Oxidative stress and low-grade chronic inflammation are tightly linked pathophysiological processes in many chronic diseases including cardiovascular diseases. The inflammatory process can also cause oxidative stress. The identification of primary abnormality is of clinical importance because the oxidative stress and inflammation are interdependent pathophysiological events [37].

Our studies, which included testing a broad spectrum of cytokines, growth factors, and oxidative stress-related biomarkers, suggest that GDF-15 and chemerin most effectively reflect the early stage of AS [21,28], and TRX1 joined this study.

There were some limitations to the study, e.g., the study groups were small, but this was due to the use of a large number of exclusion criteria for the study participants with the aim of reducing secondary effects of factors other than those related to AS on circulatory TRX1, MPO, cytokines and growth factors. 

Our study results suggest that not only pro- and anti-inflammatory biomarkers, but also TRX1 may have clinical utility in risk stratification for AS status and outcomes. The clinical value of individual biomarkers at single points in time is limited. Hence, the future of biomarker application in heart failure, including AS, lies in the multimarker panel strategy [38]. The results of our study may help to ameliorate the early diagnosis of AS and could improve the patient’s quality of life in the future.

## 5. Conclusions

AS is associated with significantly increased plasma TRX1 levels, and it is most pronounced in patients with mild AS. TRX1 might serve as a specific and sensitive biomarker of AS, especially at the mild stage. TRX1 and also chemerin, GDF-15, VEGF-A, FGF-2 and FGF-21 have significant a correlation with AS severity degrees. TRX1 also showed a positive association with FGF-2, VEGF-A, and MMP-3 in all AS patients.

Concluding Remarks: To detect AS at an early stage, we recommend monitoring plasma TRX1 levels in the relevant patient at risk.To increase the diagnostic plausibility, we also recommend the measurement of plasma GDF-15 and chemerin levels in patients with suspected AS [21,28].Screening for the above-named cytokines prior to the EchoCG may appear a cost-efficient approach in aging populations with increasing incidence of AS.A definite early stage of AS will be confirmed by performing the EchoCG examination and meeting the criteria: *Vmax* 2.5–2.9 m/s, PG < 2 0 mmHg, AVA > 1.5 cm^2^, indexed AVA > 0.85 [23,24].

## Figures and Tables

**Figure 1 biomedicines-09-01041-f001:**
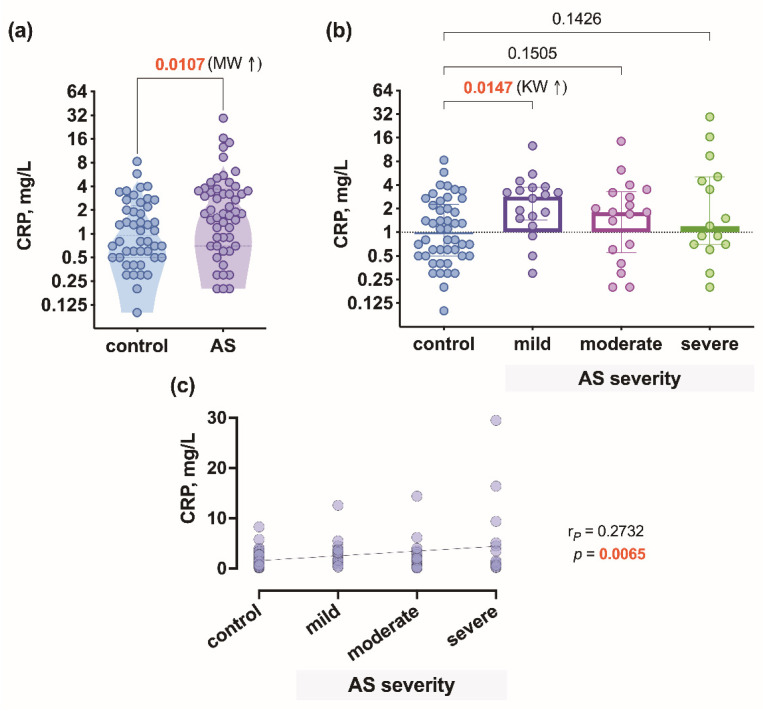
Plasma CRP level: (**a**) comparison of CRP concentration in control subjects and aortic valve stenosis (AS) patients; (**b**) comparison of CRP concentration in control subjects and AS patients with 3 different severity degrees; (**c**) correlation of CRP regarding AS severity degrees. r*_P_*—Pearson’s correlation coefficient; MW—Mann–Whitney test, KW—Kruskal–Wallis test with a post hoc procedure.

**Figure 2 biomedicines-09-01041-f002:**
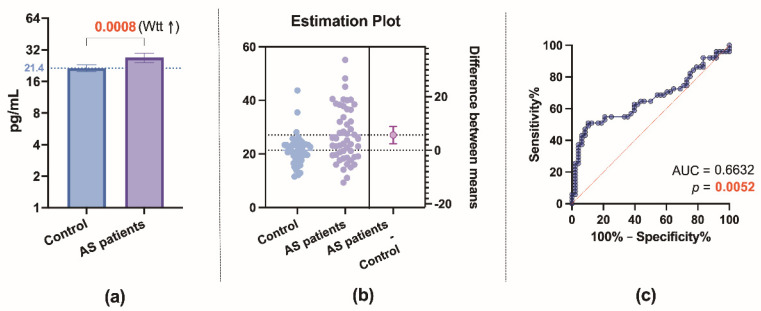
Plasma thioredoxin-1 (TRX1): (**a**) comparison of TRX1 concentration in control subjects and aortic valve stenosis (AS) patients. Wtt—Welch t test; (**b**) estimation plot displaying the raw data and the 95% confidence interval for the difference between means; (**c**) receiver-operating characteristic curve for TRX1 as a diagnostic marker of AS (control group vs. AS patient group). AUC—the area under the curve.

**Figure 3 biomedicines-09-01041-f003:**
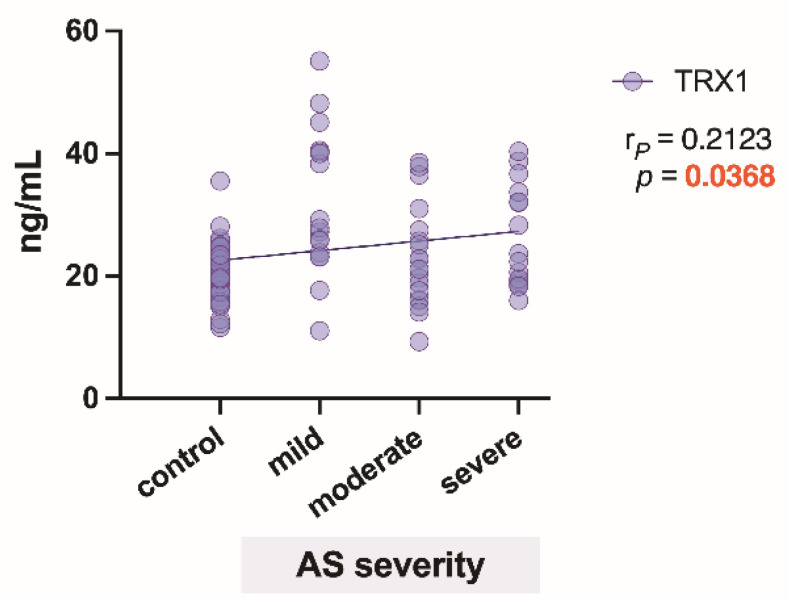
Correlation and regression line of thioredoxin-1 (TRX1) regarding AS severity degrees. r*_P_*—Pearson’s correlation coefficient.

**Figure 4 biomedicines-09-01041-f004:**
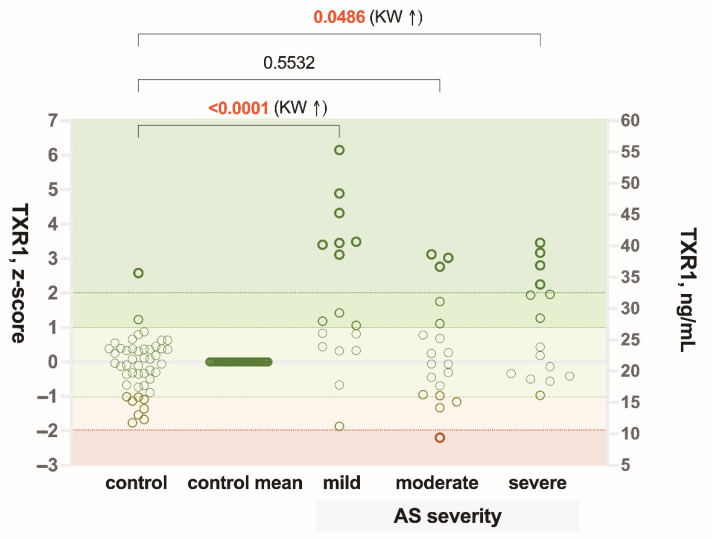
Comparison of plasma thioredoxin-1 (TRX1) concentration in mild, moderate and severe aortic valve stenosis (AS) patients to control subjects. KW—Kruskal–Wallis test; z-score—distance of data point from the control mean, expressed in standard deviations.

**Figure 5 biomedicines-09-01041-f005:**
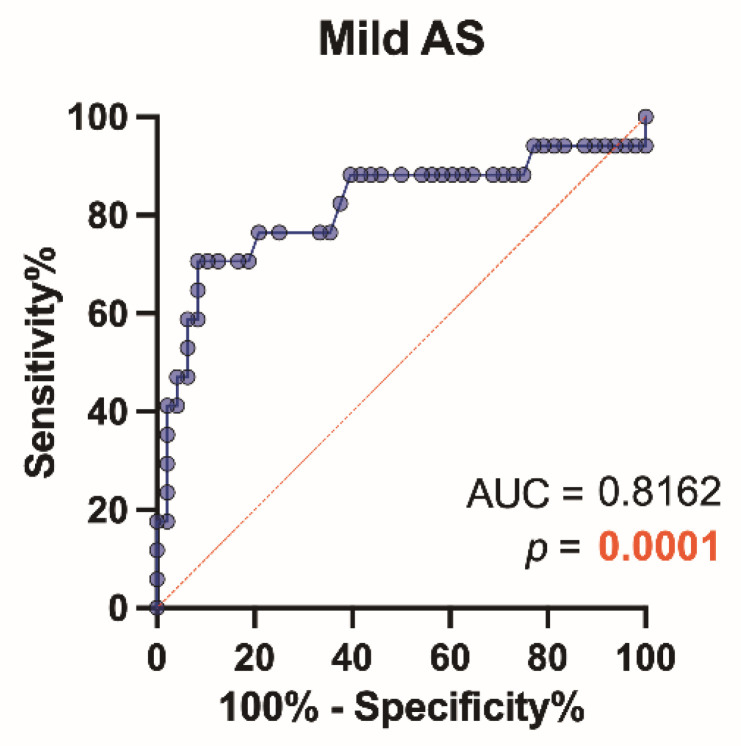
Receiver-operating characteristic curve for thioredoxin-1 (TRX1) as a diagnostic marker of AS (control group vs. mild AS patient group). AUC—the area under the curve.

**Figure 6 biomedicines-09-01041-f006:**
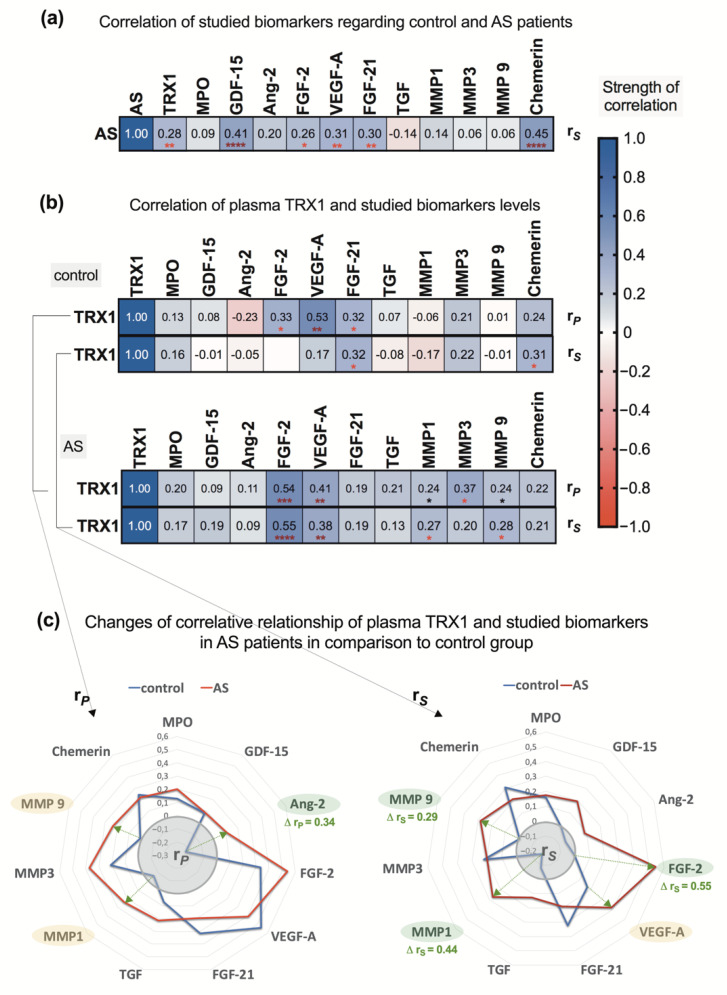
Correlative associations of the studied biomarkers: (**a**) regarding control and AS patients; (**b**) regarding TRX1 and biomarkers in control and AS patients; (**c**) comparative changes in correlative relation and respective biomarkers; r*_P_* and r*_S_*: correlation coefficients (Pearson’s and Spearman’s, respective); green arrows show the most pronounced difference between correlation coefficients of the respective biomarker, that was calculated by expression Δr = r_max_ − r_min_; closeness of associative relations is characterized by *p*-value—* 0.08 < *p* > 0.05, *
*p* < 0.05, **
*p* < 0.01, ***
*p* < 0.001, ****
*p* < 0.0001.

**Figure 7 biomedicines-09-01041-f007:**
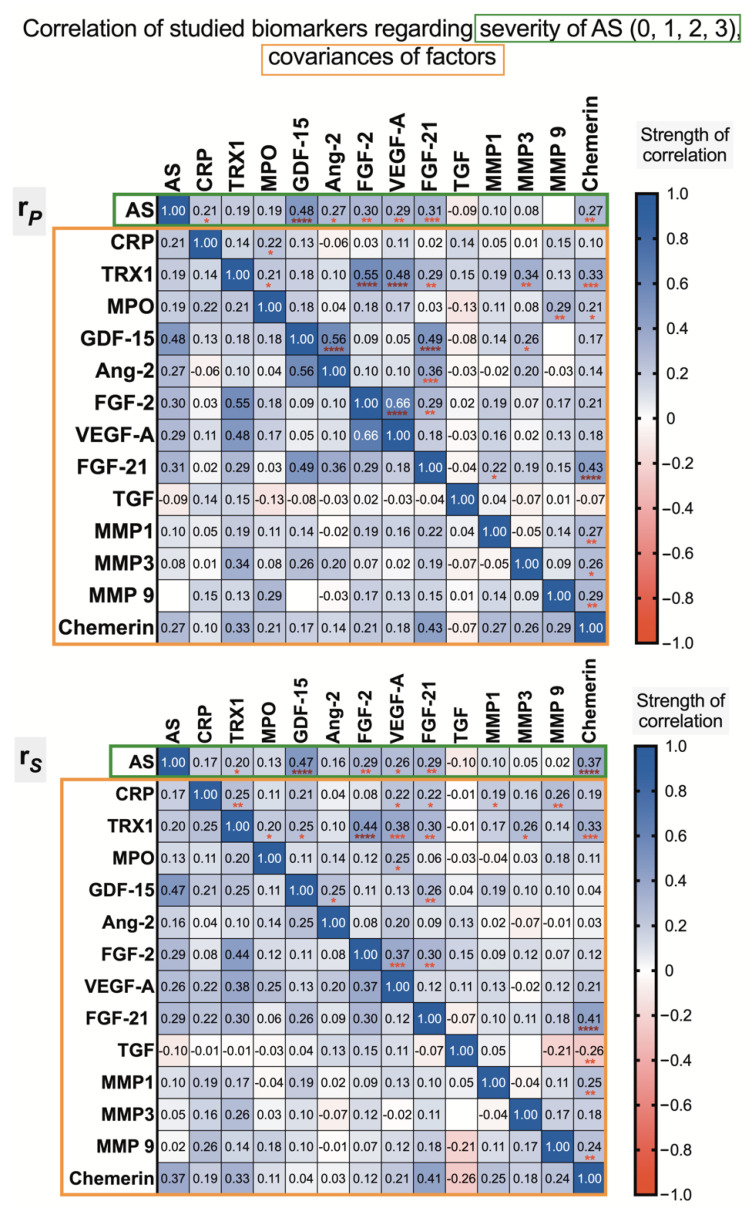
Corelation of studied biomarkers regarding severity of AS (0—no (control), 1—mild, 2—moderate, 3—severe; closeness of associative relations is characterized by *p*-value—*
*p* < 0.05, **
*p* < 0.01, ***
*p* < 0.001, ****
*p* < 0.0001; green frame); covariances of studied biomarkers (orange frame); r*_P_*—Pearson’s correlation coefficients, r*_S_*—Spearman’s rank correlation coefficients.

**Figure 8 biomedicines-09-01041-f008:**
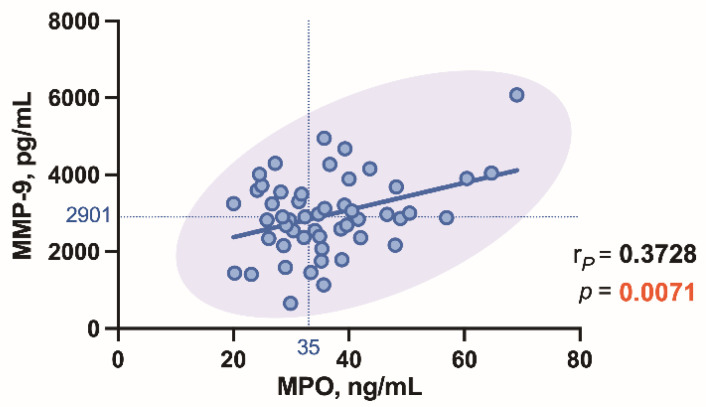
Correlation between MPO and MMP-9 levels in the aortic valve stenosis patients.

**Table 1 biomedicines-09-01041-t001:** Basic data of individuals in the control group and patients in the AS groups.

		Control (*n* = 30)	Mild Aortic Valve Stenosis (*n* = 16)	Moderate Aortic Valve Stenosis (*n* = 19)	Severe Aortic Valve Stenosis (*n* = 11)
Gender, (%)	Male Female	6 (20) 24 (80)	1 (6) 15 (94)	8 (42) 11 (58)	7 (64) 4 (36)
Age, (years)	Mdn (IQR)	70 (60–75)	72 (66–75)	74 (65–79)	69 (60–75)
BMI ^1^	M (±SD) *p* value vs. control	27.97 (±5.10)	29.53 (±4.97) *p* = 0.16	27.18 (±4.76) *p* = 0.19	27.02 (±4.04) *p* = 0.30
SV ^2^, mL	M (±SD) *p* value vs. control	81.97 (±22.20)	72.13 (±11.99) *p* = 0.26	79.89 (±20.45) *p* = 0.45	78.09 (±17.39) *p* = 0.12
EF ^3^, %	M (±SD) *p* value vs. control	61.22 (±6.44)	57.58 (±9.79) *p* = 0.17	61.32 (±8.24) *p* = 0.17	57.73 (±8.65) *p* = 0.15
SVI ^4^	M (±SD) *p* value vs. control	44.45 (±11.19)	39.26 (±8.39) *p* = 0.38	42.36 (±11.67) *p* = 0.51	41.97 (±9.99) *p* = 0.11

^1^ BMI—body mass index, weight in kilograms divided by the square of the height in meters (kg/m^2^); ^2^ SV—stroke volume, measured by left ventricular outflow method; ^3^ EF—ejection fraction, measured using Simpson’s method; ^4^ SVI—stroke volume index, the relation between the stroke volume and size of the person body surface area (mL/m^2^).

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
