# Peer review of "Thioredoxin-1 and Correlations of the Plasma Cytokines Regarding Aortic Valve Stenosis Severity"

_biomedicines, 2021, doi:10.3390/biomedicines9081041_

Round 1

Reviewer 1 Report

The current study reported Thioredoxin-1 as an important biomarker for mild Aortic valve stenosis. This study could open a new window for research and may lead to earlier disease diagnosis. However, I have some suggestion in the attachment.

Author Response

Please see our response in the attachment file 1-2021_PAPER_BIOMEDICINES-1253486_Answers to Reviewer_1

Reviewer 2 Report

This manuscript lacks the focus, the rationale and a justifiable approach. The cohort used in the study doesn’t justify the conclusion. The other major issue with the paper is the need grammatical and structural English mistakes. The manuscript in its current form doesn’t provide adequate novelty or valuable scientific knowledge.

Author Response

Please see our response in the attachment file 1-2021_PAPER_BIOMEDICINES-1253486_Answers to Reviewer_2

Reviewer 3 Report

The manuscript covers a very interesting and novel topic trying address relevant questions around the correlations of oxidative stress biomarker, TXR1 and the plasma inflammatory biomarkers regarding AS severity (mild, moderate and severe stenosis).

However, the manuscript would be improved by addressing several important points:

  • the introduction section displays a list of information. Authors should make it more homogeneous and should specify the role of each biomarker mentioned in the AS.
  • in the methods section, specify the manufacturer of the elisa kits and the relative catalog code
  • Moreover, since several studies have shown a close correlation between the levels of C-reactive protein (CRP) and the pathology studied (AS), the authors should also have evaluated the levels of this marker of systemic inflammation.
  • in the discussion section the authors should explain in more detail the significance and clinical importance of the interesting highlighted correlations between txr-1 and pro- and anti-inflammatory biomarkers.

Furthermore, for future studies, considering that the onset of AS usually begins around the age of 60, it would be interesting to evaluate whether the biomarkers analyzed could be considered predictors of this disease.

Author Response

Please see our response in the attachment file 3-2021_PAPER_BIOMEDICINES-1253486_Answers to Reviewer_3

Round 2

Reviewer 2 Report

I don’t see improvement. The points raised in relation to the first version were not address adequately. The answers provided to the points were merely confrontational and not based on scientific facts. Yes, there is some improvement to the English text. 

Author Response

Dear reviewer (no 2),

Our answers to your comments on the manuscript “I don’t see improvement. The points raised in relation to the first version were not address adequately. The answers provided to the points were merely confrontational and not based on scientific facts. Yes, there is some improvement to the English text.”:

We have made significant improvements on our manuscript, especially those pointed out by other reviewers, as they were clearly identified. In addition, the other two reviewers have accepted our edits and additions to the manuscript. Of course, we tried to take your comments into account as best we could.

Your comments on our manuscript are as follows: “This manuscript lacks the focus, the rationale and a justifiable approach. The cohort used in the study doesn’t justify the conclusion. The other major issue with the paper is the need grammatical and structural English mistakes. The manuscript in its current form doesn’t provide adequate novelty or valuable scientific knowledge.”

In this manuscript, we hypothesized that TRX1 in AS patients is increased in plasma and correlates with the severity of AS, and TRX1 may serve as a clinical biomarker. This hypothesis was confirmed. Patients were recruited in the study according to strict inclusion and exclusion criteria, using appropriate research methods, including a high-level statistical analysis of the data obtained. Please, allow us to repeat from the previous answer - our manuscript demonstrates for the first time the results that TRX1 might serve as specific and sensitive biomarker of AS. In addition, our manuscript expands knowledge about the relationship of TRX1 to other cytokines and growth factors.

Reviewer 3 Report

Dear authors, thank you for the review. The manuscript "Thioredoxin-1 and correlations of plasma cytokines concerning the severity of aortic valve stenosis" has certainly been improved in all sections. Sincerely

Author Response

Dear reviewer (no 3),

Thank you very much for your review.